# Bumblebees retrieve only the ordinal ranking of foraging options when comparing memories obtained in distinct settings

Cwyn Solvi[1,2†], Yonghe Zhou[1,3†], Yunxiao Feng[1], Yuyi Lu[1], Mark Roper[3], Li Sun[1], Rebecca J Reid[3], Lars Chittka[3], Andrew B Barron[4], Fei Peng[1,5,6]*

[1]Department of Psychology, School of Public Health, Southern Medical University, Guangzhou, China; [2]Ecology and Genetics Research Unit, University of Oulu, Oulu, Finland; [3]Biological and Experimental Psychology, School of Biological and Behavioural Sciences, Queen Mary University of London, London, United Kingdom; [4]Department of Biological Sciences, Macquarie University, Sydney, Australia; [5]Department of Psychiatry, Zhujiang Hospital, Southern Medical University, Guangzhou, China; [6]Guangdong-Hong Kong-Macao Greater Bay Area Center for Brain Science and Brain-Inspired Intelligence, Southern Medical University, Guangzhou, China

*For correspondence:
fpeng@smu.edu.cn

†These authors contributed equally to this work

Competing interest: The authors declare that no competing interests exist.

**Abstract** Are animals' preferences determined by absolute memories for options (e.g. reward sizes) or by their remembered ranking (better/worse)? The only studies examining this question suggest humans and starlings utilise memories for both absolute and relative information. We show that bumblebees' learned preferences are based only on memories of ordinal comparisons. A series of experiments showed that after learning to discriminate pairs of different flowers by sucrose concentration, bumblebees preferred flowers (in novel pairings) with (1) higher ranking over equal absolute reward, (2) higher ranking over higher absolute reward, and (3) identical qualitative ranking but different quantitative ranking equally. Bumblebees used absolute information in order to rank different flowers. However, additional experiments revealed that, even when ranking information was absent (i.e. bees learned one flower at a time), memories for absolute information were lost or could no longer be retrieved after at most 1 hr. Our results illuminate a divergent mechanism for bees (compared to starlings and humans) of learned preferences that may have arisen from different adaptations to their natural environment.

## Editor's evaluation

This is a very informative and nicely controlled study showing that, when retrieving the value of a food type, bumblebees guide their choices by remembered ranking of feeders instead of focusing on their absolute rewards.

## Introduction

What do animals remember about items out of context? For example, suppose we learn that different options (e.g. coffee shops) result in different reward outcomes (e.g. waiting time and quality), and later we are presented with a choice between two previously encountered options which we have never

experienced side-by-side. What types of values do we remember for those options now presented in a novel context? Do our memories of the subjective values for each option contain absolute information (e.g. delay to reward), remembered ranking (how they compared to previous alternatives), or a weighted combination of both?

In typical studies exploring the economic choices of animals including humans, subjects do not have to use distant memories of the options; they are presented with choices where the objective values (e.g. amount, cost, and status) are concurrently visible and can be directly compared. Under such conditions, a wealth of research shows that animals' choices can be influenced by the presence of additional options (*Hunter and Daw, 2021*; *Spektor et al., 2021*). An example of this phenomenon is frequently used in marketing: when given a choice between popcorn options in different sizes, e.g. $3 for small and $7 for large, most people choose the smaller cheaper option, but when a $6 medium option is added, more people choose the large because it now seems like a good deal. Evidence of contextual effects like this on direct assessments has been found across the animal kingdom, e.g. humans and other primates (*Berkowitsch et al., 2014*; *Parrish et al., 2015*; *Trueblood et al., 2013*), bats (*Hemingway et al., 2021*), birds (*Bateson, 2002*; *Morgan et al., 2012*), frogs (*Lea and Ryan, 2015*), fish (*Reding and Cummings, 2017*), bees (*Shafir et al., 2002*), and worms (*Iwanir et al., 2019*). However, little is known about the type and degree of information (absolute and/or relative) that is encoded in the remembered subjective values of options.

Only more recently have investigations of absolute and relative information traversed into the realm of reinforcement learning, where value must be inferred from memories. Studies on starlings (*Pompilio and Kacelnik, 2010*) and humans (*Bavard et al., 2018*; *Bavard et al., 2021*; *Klein et al., 2017*) demonstrated that both absolute memories and remembered ranking are combined in particular ways to give rise to these animals' preferences. So far, however, no other species have been investigated for the roles played by absolute memories and remembered ranking in learned preferences. Here, we examine this in bumblebees (*Bombus terrestris*), an invertebrate and a key model for examining the economy of decision-making outside of humans (*Real, 1996*).

Specifically, we adopt an instrumental learning paradigm that combines a contextual training phase and a transfer-test phase (*Palminteri and Lebreton, 2021*). This paradigm essentially involves two distinct learning contexts (e.g. AB context and CD context; *Figure 1*), with each context offering two options of contrasting properties (e.g. A>B and C>D in reward sizes). After training, animals' learned preferences are tested with a novel combination of options (e.g. B vs C). Note that behavioural tests for transitive inference (*Guez et al., 2013*; *Vasconcelos, 2008*) involve a similar task design, which includes several training phases and a test phase of new combinations. However, this method provides overlapped relational premises during training (e.g. A>B, B>C, C>D, and D>E) in order to examine if animals can infer the relationship between a pair of options within the chained sequence which had previously not been experienced together (e.g. B vs D). In contrast, our paradigm provides no direct overlapped training between contexts (e.g. A>B and C>D), and therefore, animals cannot infer the relationship between unchained options (e.g. B vs C). Rather, by providing both absolute and relative information during training, our paradigm is used to assess whether, and in what combination, bumblebees retain and utilise absolute and ranking memories.

## Results

If bumblebees encode and retrieve memories for absolute values, their preference for a particular option should not depend on the context in which that option was learned (*Padoa-Schioppa and Assad, 2008*). To test this idea, we conducted experiment 1 with a multi-contextual design where certain flowers had the same quality of reward but different ranking. Bees were first trained (individually in all experiments) on two different pairs of coloured flowers (A and B; C and D; *Figure 1A*, *Figure 1—figure supplement 1*, and *Figure 1—figure supplement 2*). The sucrose concentrations of the different pairs of flowers were chosen according to Weber's Law (*Akre and Johnsen, 2014*) to represent the same perceived difference, i.e. A:B=45%:30% had the same contrast of incentives as C:D=30%:20% (*Figure 1A*). Note that for all experiments, the order of training sessions was counterbalanced to ensure that test preferences were not a result of a recency effect. Meanwhile, the focal colours used in the unrewarded test were counterbalanced during training to account for any colour effects (Materials and methods; *Figure 1*; *Figure 1—figure supplement 3*). Following sequential training in both contexts (*Figure 1—figure supplement 3* and *Figure 1—figure supplement 4*), we

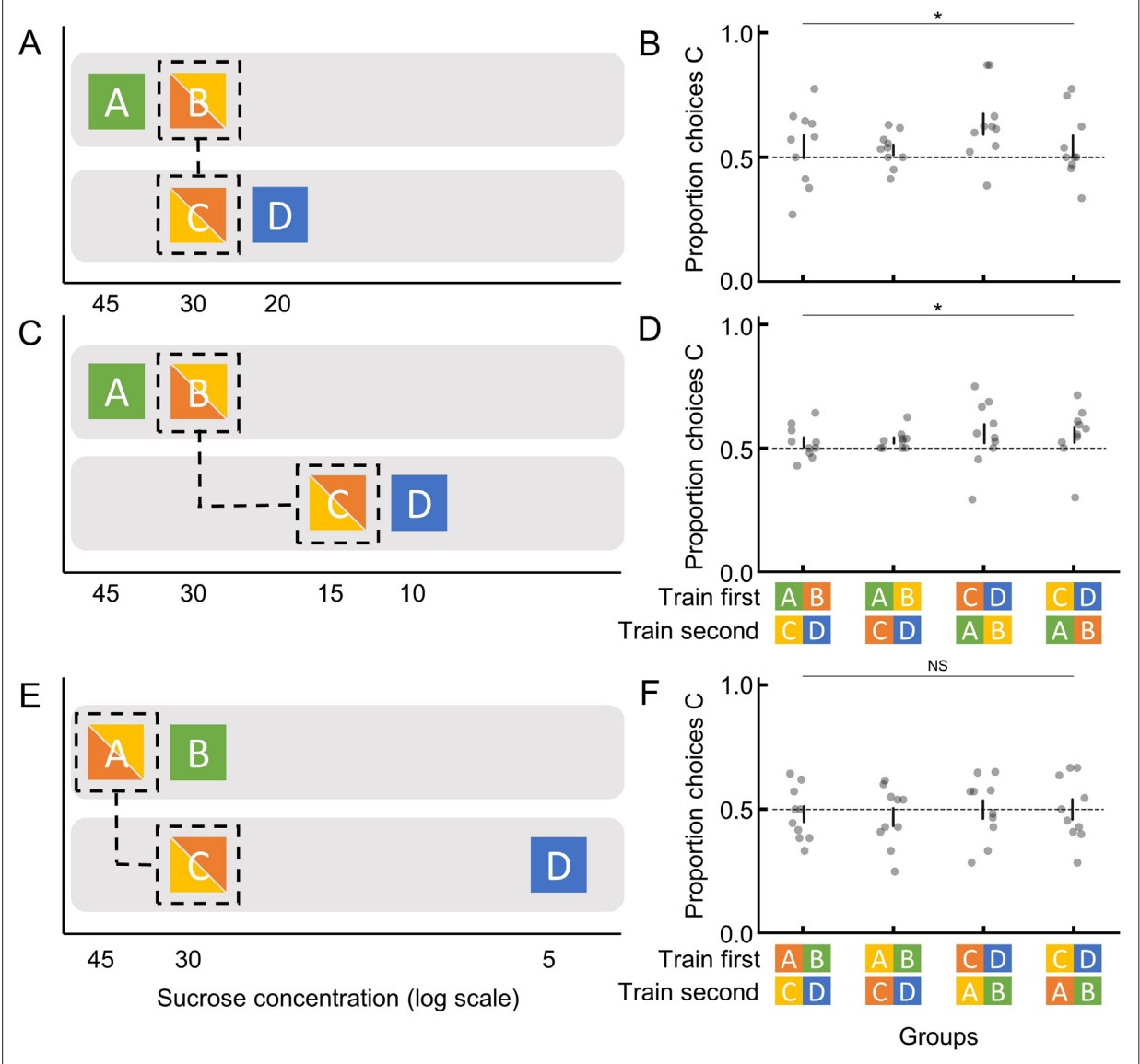

**Figure 1.** Bumblebees make decisions based on ordinal comparisons. (**A**, **C**, and **E**) The corresponding sucrose concentration of each stimulus is displayed on a log scale to visually represent their relative differences according to Weber's Law (**Akre and Johnsen, 2014**). Training sessions are indicated by separate grey backgrounds, and the test options in each experiment are indicated with dashed lines. The bi-colour squares indicate that the colours for the focal options used were counterbalanced across bees. (**B**, **D**, and **F**) Test results for each experiment. Groups indicate different counterbalanced training sequence and colour-reward contingency (see **Figure 1—figure supplement 3** for more details). Each filled circle represents the proportion of choices for option C by an individual bumblebee (10 individuals per group). Dashed horizontal lines indicate chance performance. Vertical lines indicate mean ± SEM. p values were calculated from generalised linear models (Materials and methods); NS: p > 0.05; *: p < 0.01.

The online version of this article includes the following figure supplement(s) for figure 1:

**Figure supplement 1.** General setup for experiments 1, 2, and 3.

**Figure supplement 2.** Specifications of the colours used for all the experiments.

**Figure supplement 3.** Counterbalanced colour sets used in experiments 1–6.

**Figure supplement 4.** Bees can discriminate flowers of different colours and sucrose concentrations.

tested each bee's preference between flower types B and C, which had the same reward quality (30%) during training. Therefore, if bumblebees' preferences were driven exclusively by their memory for absolute values for options, they should show no preference between B and C in the test. Any significant preference would suggest a contextual effect due to B and C having different rankings during training (B<A while C>D). In an unrewarding test with B and C flowers, bees significantly preferred C

**Table 1.** Predictions for various decision strategies for bumblebees' flower preferences in experiments 1, 2, and 3.

The left column lists the different categories of decision strategies. The middle three columns show the predicted results of each strategy for each experiment. The right column shows whether the predictions of each strategy match the behaviour of bumblebees in all experiments. A lexicographic combination strategy is where choices are determined by a difference in a priority dimension, but if options are equal in that dimension, then choices rely on a secondary dimension. For example, bumblebees might choose between two flower types based on a remembered ranking (as long as flowers differed in ranking during training), no matter what the absolute sugar concentration differences were. However, if the remembered rankings of the flowers are the same (i.e. they cannot use ranking to make a choice), bumblebees might then use any difference in absolute memory. In contrast, a non-lexicographic combination strategy is where differences in either dimension can be used to make a choice. For example, bumblebees might use absolute memories to choose flowers that differ in ranking, even if the ranking and absolute contrasts are distinct (e.g. one ordinal rank difference vs threefold absolute difference). Value by association entails one option acquiring a higher value because it was experienced in a richer environment (*Pompilio and Kacelnik, 2010*), i.e. bumblebees might assign a higher value to flower B because it was experienced with A (the highest rewarding flower), compared to a lower value assigned to C because it was experienced with D (the lowest rewarding flower). State-dependent valuation learning (SDVL) refers to assigning values to options based on whether subjects were in a better or worse internal state when experiencing those options (*McNamara et al., 2012*). For example, bumblebees may assign different values to a flower type based on whether they are foraging in a more or less profitable environment, i.e. based on the comparison between each flower type's sugar concentration and the average sucrose concentration within the bee's current crop load.

| | Expected preference | | | |
| --- | --- | --- | --- | --- |
| Strategy | Experiment 1 **B (30%) vs C (30%)** | Experiment 2 **B (30%) vs C (15%)** | Experiment 3 **A (45%) vs C (30%)** | Matches bees' behaviour |
| Absolute memory | Indifferent | B | A | ✗ |
| Remembered ranking | C | C | Indifferent | ✓ |
| Lexicographic combination—absolute memory priority | C | B | A | ✗ |
| Lexicographic combination—ranking priority | C | C | A | ✗ |
| Non-lexicographic combination | C | B/C/indifferent | A | ✗ |
| Value by association | B | B | A | ✗ |
| State-dependent valuation learning | C | C | C | ✗ |

over B (generalised linear model [GLM]: 95% CI = [0.08, 0.38], N=40, and p=4.00e-3; *Figure 1B* and *Figure 1—figure supplement 3*). Although these results suggest that bees store memories for the relative ranking of flowers, they still may encode absolute information (*Table 1*), and therefore, we carried out additional experiments.

Bees may still store and use memories for both absolute and relative information in some combined fashion. To test this, in experiment 2, we set the reward quality of flowers B and C to be different (*Figure 1C* and *Figure 1—figure supplement 3*). The sucrose concentration of flowers was A>B >> C>D (45%, 30%, 15%, and 10%). After training in both contexts, bees' preference between B and C flowers was tested. If bumblebees retain and use memories for both ranking and absolute information (and the value weights for these two different types of information are not extremely unbalanced), bees should either prefer B over C because it is much higher in quality, or this value difference would cancel out the ranking difference (B<A while C>D), and their preference for B and C should be equal. On the other hand, a preference for the much lower reward quality flower C would indicate they have

access to only ranking memories. Similar to experiment 1, the result of the unrewarding test showed that bees preferred C (GLM: 95% CI = [0.07, 0.28], N=40, and p=3.30e-3; *Figure 1D*).

Note that a preference for C over B in experiments 1 and 2 may have resulted from remembered ranking or a distorted absolute memory for flower type B due to the presence of and comparison with A during training. Evidence for the latter would manifest as a difference in the preference for C over B in experiment 2 compared to in experiment 1, whereas remembered ranking would predict a similar preference. There was no significant difference in preference for C across experiments 1 and 2 (GLM: 95% CI = [–0.24, 0.13], N=80, and p=0.55; *Figure 1B and D*). There is thus no evidence that bumblebees' choices in novel contexts are based on remembered absolute metrics.

If bumblebees rely purely on remembered ranking, then their memories for options should only be ordinal (*Vlaev et al., 2011*). That is, they can tell one option is better than another but cannot tell how much better. If so, there should be no indication that bumblebees can compare options in quantitative terms. To test this, in experiment 3, we made the reward contrast ratios within the two contexts very different ([A:B] = [1.5:1]=45%:30% and [C:D] = [6:1]=30%:5%; A>B = C>>D; *Figure 1E*). Here, testing bees' preference between A and C flowers can answer our question. If bees encode memories for relative differences in quantitative terms, C should be preferred over A, as C had a higher relative value than A. Alternatively, if bees only remember the ordinal rank of options, preference for A and C should be the same because they have the same rank within their training contexts. In the unrewarding test, bees chose A and C equally (GLM: 95% CI = [–0.16, 0.12], N=40, and p=0.79; *Figure 1F*), suggesting bumblebees store and recall only memories for the qualitative difference between options. Furthermore, bumblebees' lack of preference for the equally ranked A and C flowers, despite their difference in sugar concentration, demonstrates absolute memories were not remembered (*Table 1*) and, in combination with experiments 1 and 2, suggests that bumblebees only encode and recall remembered ranking for use in novel contexts.

It may be that bumblebees can store and recall absolute information, but any ranking information available overshadows absolute memory. To test this, in experiment 4, we trained bees so that ranking information was absent. Bees experienced a temporally separated training procedure (Materials and methods; *Figure 2A*) whereby they first foraged on one flower with either 45% or 30% sugar solution, and 1 hr later they were trained on a different flower with either 30% or 45%, respectively. In the subsequent (1 hr later) unrewarded test, bees' preferences were assessed. Because the two flower types were separated physically and temporally, bees had no ranking information available to putatively overshadow absolute information. Therefore, any preferences formed would be a result of remembered absolute information. However, bees had no significant preferences for either option (GLM: 95% CI = [–0.15, 0.22], N=20, and p=0.69; *Figure 2B*), suggesting that bees' lack of absolute memory in the tests of experiments 1–3 was unlikely to be a result of absolute memories being less salient and overshadowed by ranking memories.

In order to discriminate any two sequentially visited flowers and indeed to rank them, bumblebees must utilise absolute information for at least a brief amount of time. During training in experiments 1–3, bumblebees could see flowers of both types simultaneously. Given the results of experiment 4, we therefore asked whether bumblebees could utilise absolute information long enough to rank flowers that were temporally close together but visually separated. To do this, in experiment 5, we trained bees on one flower for each bout, i.e. each visit to the arena (Materials and methods; *Figure 2C*). This way, both flower types were never in a bee's visual field at the same time. During the unrewarded test, bees preferred the option that had offered the higher sugar concentration in training (GLM: 95% CI = [0.23, 0.87], N=10, and p=8.29e-3; *Figure 2D*), suggesting that absolute information about a flower learned in visual isolation can be retained for at least a couple of minutes but, as experiment 4 highlights, is eventually lost or not able to be recalled later. Experiment 4 and 5 thus collectively indicate that bumblebees can use absolute information to compare options only in their short-term memories (range of minutes) but not in their mid-term memories (range of hours; *Menzel, 2001*).

Our results highlight the potential disadvantage of using only remembered ranking, i.e. preferring to search for a less rewarding flower. How might our results speak to bumblebees' behaviour in more 'natural' settings? In the wild, bumblebees often forage in situations whereby when drinking from one flower type they cannot see other flower types in their field of view, i.e. on the other side of a bush or many meters away. Yet, the distance between any two flowers within a bumblebee's foraging range can be traversed within minutes (*Osborne et al., 1999*), i.e. within the range of bumblebees'

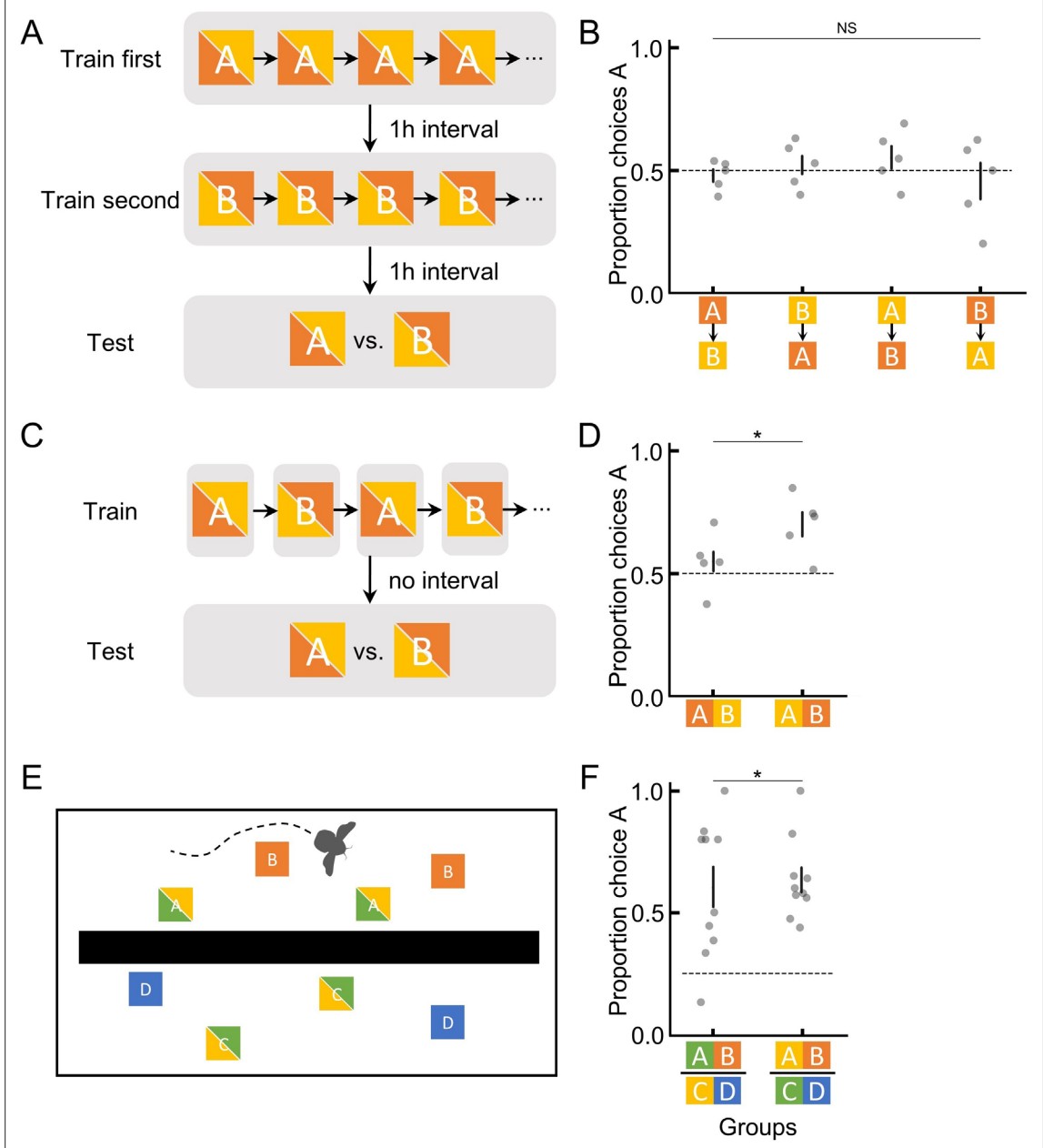

**Figure 2.** Bumblebees are only able to utilise absolute information for short periods of time in order to encode and recall ranking information in novel contexts. (**A**) Illustration of the training and testing procedure in experiment 4, where two options were separately trained and tested, with a 1-hr interval in between to ensure clear spatial and temporal separation (and no ranking information). (**C**) The training and testing procedure in experiment 5, where two options were trained in alternating bouts (spatially separated but temporally close). (**E**) Top view of setup for experiment 6 where a wall separated flowers such that the bee could not see both groups of flowers at the same time. (**B**, **D**, and **F**) Results of the unrewarding test in experiments 4–6. Groups indicate different colour-reward contingency for bees (B & D: 5 bees per group; F: 10 bees per group; *Figure 1—figure supplement 3*). The bi-colour squares indicate that the colours for the focal options used were counterbalanced across bees. Dashed horizontal lines indicate chance performance. Vertical lines indicate mean ± SEM. p values were calculated from generalised linear models (Materials and methods); NS: p > 0.05; *: p < 0.01.

short-term memories (*Menzel, 2001*). Therefore, we would expect that bumblebees in the wild would not be susceptible to learning preferences for a less rewarding flower despite being unable to retain absolute information for very long. In an attempt to formally test this idea, in experiment 6, we trained bees in a multi-contextual semi-realistic foraging situation. Here, in a divided arena, bumblebees learned to forage from flowers of four different colours (*Figure 2E* and *Figure 1—figure supplement*

*3*) but could only see flowers of two colours at any one time, i.e. could only see A and B (45% and 30%) flowers from one side of the arena and could only see C and D (30% and 20%) flowers from the other side (*Figure 2E*). In the unrewarding test with all four options available, bees preferred option A (GLM: 95% CI = [0.94, 1.68], N=20, and p=1.21e-6; *Figure 2F*), indicating that the short retention/ utilisation of absolute memories that bumblebees showed in our lab-created situations are likely not to be disadvantageous to bumblebees in the wild, given their natural behaviour and ecological niche.

## Discussion

Our results suggest that bumblebees are only able to make use of ordinal ranking memories to guide foraging choices outside their original learning contexts. Although absolute information is essential for any animal to compare any two options initially, our findings show that only ordinal ranking can be recalled by bumblebees later in new contexts. Our results suggest that temporal adjacency is necessary for absolute information of different options to be compared by bees. That is, once bees experience one option, in order to utilise the absolute information, they must experience a second option within several minutes. Both our proof-of-concept experiment (experiment 6) and other previous multi-choice semi-realistic foraging tasks (e.g. *Greggers and Menzel, 1993*) show that with short inter-flower visits, bees can utilise absolute information in order to compare and rank flowers.

Another potential mechanism worth discussing which could potentially explain bumblebees' preferences is state-dependent valuation learning (SDVL; for a review see *McNamara et al., 2012*; *Table 1*). If bees' choices were a result of SDVL instead of remembered ordinal ranking, then their preferences should be quantitative, i.e. larger values should be assigned with larger differences between internal state and flower experience. However, the results of experiment 3 speak against this interpretation because bees were indifferent between equally ranked A and C even though they were experienced in a richer and poorer environment, respectively.

Why would bumblebees have evolved to use only memories for ordinal comparisons while humans and starlings evolved to retain and recall both absolute and ranking memories? Breadth of diet has been suggested to play a role in the evolution of cognition (*Hemingway et al., 2017*; *MacLean et al., 2014*; *Simons and Tibbetts, 2019*). Humans and starlings forage on a range of different foods, whereas adult bumblebees feed almost exclusively on nectar and pollen from flowers. Perhaps a varied diet, whereby one might need a common currency across vastly different food types (*Chib et al., 2009*; *Levy and Glimcher, 2012*) may have forced some animals to retain and use absolute memories, whereas a limited diet and limited dimensionality of reward might have favoured memories for ordinal relationships alone.

There is no reason yet to suggest a bee's brain lacks the neural substrates to retain and recall absolute memories for options for later use in novel contexts. In fact, recent electrophysiological recordings on the gustatory neurons of bumblebees demonstrated that the spiking rates of these peripheral sensory neurons increased as a function of sucrose concentration (*Miriyala et al., 2018*). These gustatory neuronal signals could theoretically be used by higher centres of an animal's brain to encode an option's utility as a cardinal (absolute) value (*Schultz, 2015*). However, our results suggest that bumblebees do not access or utilise any potentially stored absolute memories when it would be of benefit in novel contexts. Given our results, it would be intriguing to determine the differences in the underlying neural mechanisms responsible for the utilisation of absolute properties and ordinal ranking memories. Whatever the ultimate and proximate causes of bumblebees' ordinal-only memories for options' values, our findings demonstrate a fundamental difference in the mechanisms underlying learned preferences between those of bumblebees and those of starlings and humans.

## Materials and methods
### Animals and setup

Bumblebee (*Bombus terrestris*) colonies were obtained from the Chinese branch of the Biobest Group (Biobest Belgium N.V., Westerlo, Belgium) and housed in wooden nest boxes (28 cm × 16 cm × 11 cm). Although there are no current requirements regarding insect care and use in research, experimental design and procedures were guided by the 3Rs principles (*replacement*, *reduction*, and *refinement*; *Russell and Burch, 1959*). The behavioural tests were non-invasive, and the types of manipulations

used (sucrose and water) are all experienced by bumblebees during their natural foraging life in the wild. The bumblebees were cared for on a daily basis by trained and competent staff, which included routine monitoring of welfare and provision of correct and adequate food during the experimental period. A foraging arena (40 cm × 59 cm × 41 cm; *Figure 1—figure supplement 1*) was connected to the nest boxes via an acrylic tunnel with sliding doors, allowing experimenters to control bees' access to the arena. Individual bees were marked with number tags (Opalithplättchen, Warnholz & Bienen-voigt, Ellerau, Germany), which were super-glued to the bees' thorax. To ensure the sucrose solution concentration used in experiments was motivating, colonies were fed outside of experiments with 5–15% sucrose solution (w/w). They were also provided with ~3 g pollen every day. Illumination was provided by daylight fluorescent tubes (MASTER TL-D 90 DeLuxe 36 W/965, Philips, Eindhoven, the Netherlands) and near-UV fluorescent tubes (TL-D 36 W BLB, Philips) with high-frequency electronic ballasts (EB-Ci 1–2 36 W/1–4 18 W, 42–60 kHz, Philips) to generate a flicker frequency beyond the bumblebee's flicker-fusion frequency. Coloured acrylic squares (25 mm × 5 mm × 5 mm) set on top of opaque glass cylinders as (artificial) flowers were placed in the arena with a different random spatial arrangement each trial. The spectral reflectance (*Figure 1—figure supplement 2A*) of all flower colours used in experiments was measured with a wavelength range of 300–700 nm and with 1 nm increments, using a spectrophotometer (Ocean Optics USB 2000+; Shanghai, China) and a deute-rium/halogen light source. The perceptual positions of the colours in the bee colour hexagon space (*Figure 1—figure supplement 2B*) were calculated using the spectral reflectance measurements and the published *Bombus terrestris* spectral sensitivity functions of their three photoreceptors (*Chittka, 1992*; *Skorupski et al., 2007*). The minimal perceptual distance of all the colour combinations used in this study was above 0.14 hexagon units, which can be well differentiated by bumblebees (*Dyer and Chittka, 2004*). Furthermore, neurophysiological (*Miriyala et al., 2018*) and behavioural evidence (*Whitney et al., 2008*) verify that the sugar concentration differences used in this study can be readily discriminated by bumblebees.

## Experimental protocol

All bees were individually pre-trained on eight transparent artificial flowers and were allowed to collect a full crop of sucrose solution from those flowers. Once a bee had successfully foraged for at least three consecutive bouts, she was moved to the training phase of an experiment.

### Experiments 1, 2, and 3

Bees (N=40 for each experiment) were trained to forage from eight flowers, four of each of two colours (*Figure 1—figure supplement 1* and *Figure 1—figure supplement 3*). In experiment 1, one group of bees (n=10) learned (individually) that green flowers contained 45% sucrose solution and that yellow flowers contained 30% sucrose solution. Once training on these two flower types was complete (more detail below), this group of bees learned (individually) that orange flowers contained 30% sucrose solution and that blue flowers contained 20% sucrose solution. Training for bees in experiments 2 and 3 was the same except the sucrose concentrations were different. Training sequences and colour combinations were equally counterbalanced across bees in each experiment (*Figure 1—figure supplement 3*). Flower colours and sequences for all experiments are listed in *Figure 1—figure supplement 3*. Initially during training, all bees would land and drink from both flowers. As bees learned the colour-reward contingency, they began to land on the less rewarded flowers but did not drink. Once this behaviour was observed, during subsequent bouts, we removed and replaced one higher-ranking flower with one lower-ranking flower. By doing so, and because none of the flowers were refilled during training bouts, bees would again visit the lower-ranking flowers and drink the sugar water droplet. This method helped us to ensure that bees experienced drinking sucrose solution from both options an equal number of times. Training was completed when a bee collected 50 of the 20 μl aliquots of sugar reward from flowers of both colours. Note that we verified, in our setup, that bees could learn to discriminate between flowers of different colours and sucrose concentrations (*Figure 1—figure supplement 4*). After training, each bee was individually tested with two types of flowers (four of each type) offering 20 μl droplets of unrewarding water, and all flower visits were recorded for 2 min. Note that in all tests, the two flowers types presented had not been experienced previously together in the same arena by the bees.

### Experiment 4

Bees (N=20) were trained in the same way as the other experiments except that during training only one flower type was in the arena at any one time, i.e. either all of the flowers offered 45% sucrose solution or all flowers offered 30% sucrose solution (*Figure 1—figure supplement 3*; *Figure 2A*). Also, the inter-session interval for experiment 4 was 1 hr. Each session ended when the bee completed 50 flower visits. After training and after another 1 hr interval, bees underwent an unrewarded test with the two flower types they had been trained on, presented simultaneously in the arena. Both the colour/sucrose concentration contingency and the order of training sessions (which type of flower was trained last) were counterbalanced across bees (*Figure 1—figure supplement 3*; *Figure 2B*).

### Experiment 5

Bees (n=10) were trained similarly to experiment 4 in which only one flower type was in the arena at any one time (*Figure 1—figure supplement 3*; *Figure 2C*). However, instead of two long sessions, bees experienced each flower in alternating bouts (visits to the arena to fill their crop). Once bees experienced 50 flower visits on each flower type, training ended and the bees underwent an unrewarded preference test with both flower types in the arena (*Figure 1—figure supplement 3*; *Figure 2D*).

### Experiment 6

Bees (N=20) were trained and tested similarly to experiments 1–3 except that the two sets of flowers were in the arena at the same time and accessible to the bee during each bout. However, flowers were also separated by an opaque wall to prevent the bees from seeing both sets simultaneously (*Figure 1—figure supplement 3*; *Figure 2E*). The wall extended to cover most but not the full length of the arena, and there was space at the entrance and the far end of the arena so that bees could freely choose to visit flowers on either side of the wall. No matter where the bee was in the arena, she could not see both pairs of flowers at the same time. In addition, flowers were refilled after the bees emptied one and began drinking on another. Each group (n=10) was trained with either the yellow or green flower as the high concentration flower. The training was deemed complete when either a bee performed 200 landings or when a bee only landed exclusively on any one option for three consecutive bouts (note that bees never landed exclusively on the lower rewarding option for three bouts consecutively). After training, the bees were individually tested with all four flowers (two of each type) presented in the same arena with the opaque barrier removed (*Figure 1—figure supplement 3*; *Figure 2F*).

## Statistical analyses

R v.3.6.1 was used to perform all GLMs with quasibinomial distribution and logit link function. For all models, the response variable was the proportion of choices for the flower that had been associated with high concentration sugar water. Fixed factors for models for results of experiments were (i) colony of each bee, (ii) flower colour, and (iii) training sequence (whether the higher concentration sugar reward was used during the first or second training session). Significance of fixed effects was tested using likelihood ratio tests, and none were found to have a significant effect on bees' preferences during the tests in any experiment. When comparing preference for flower C across experiments 1 and 2, experiment was set as a fixed factor.

## Additional information

### Funding

| Funder | Grant reference number | Author |
|---|---|---|
| National Natural Science Foundation of China | 31700988 | Fei Peng |
| National Natural Science Foundation of China | 31970994 | Fei Peng |

| Funder | Grant reference number | Author |
|---|---|---|
| Key-Area Research and Development Program of Guangdong Province, China | 2018B030340001 | Fei Peng |
| Templeton World Charity Foundation | TWCF-2020-0539 | Andrew B Barron |
| China Scholarship Council | 202008440515 | Yonghe Zhou |

The funders had no role in study design, data collection and interpretation, or the decision to submit the work for publication.

## Author contributions

Cwyn Solvi, Conceptualization, Software, Formal analysis, Supervision, Validation, Visualization, Methodology, Writing – original draft, Writing – review and editing; Yonghe Zhou, Data curation, Software, Investigation, Visualization, Writing – original draft, Writing – review and editing; Yunxiao Feng, Li Sun, Rebecca J Reid, Investigation; Yuyi Lu, Data curation, Formal analysis; Mark Roper, Conceptualization, Investigation, Writing – review and editing; Lars Chittka, Conceptualization, Writing – original draft, Writing – review and editing; Andrew B Barron, Funding acquisition, Writing – review and editing; Fei Peng, Conceptualization, Resources, Data curation, Software, Formal analysis, Supervision, Funding acquisition, Validation, Investigation, Visualization, Methodology, Writing – original draft, Project administration, Writing – review and editing

## Author ORCIDs

Cwyn Solvi ⓘ http://orcid.org/0000-0003-2517-6179
Yonghe Zhou ⓘ http://orcid.org/0000-0003-3324-9972
Yuyi Lu ⓘ http://orcid.org/0000-0002-1635-7907
Mark Roper ⓘ http://orcid.org/0000-0003-1135-6187
Lars Chittka ⓘ http://orcid.org/0000-0001-8153-1732
Andrew B Barron ⓘ http://orcid.org/0000-0002-8135-6628
Fei Peng ⓘ http://orcid.org/0000-0002-1637-5611

## Decision letter and Author response

Decision letter https://doi.org/10.7554/eLife.78525.sa1
Author response https://doi.org/10.7554/eLife.78525.sa2

## Additional files

### Supplementary files
• MDAR checklist

### Data availability

All data generated or analysed during this study are available on the Dryad Digital Repository: https://doi.org/10.5061/dryad.rr4xgxdb9.

The following dataset was generated:

| Author(s) | Year | Dataset title | Dataset URL | Database and Identifier |
|---|---|---|---|---|
| Solvi C, Zhou Y, Feng Y, Lu Y, Roper M, Sun L, Reid R, Chittka L, Barron A, Peng F | 2022 | Bumblebees retrieve only the ordinal ranking of foraging options when comparing memories obtained in distinct settings | https://doi.org/10.5061/dryad.rr4xgxdb9 | Dryad Digital Repository, 10.5061/dryad.rr4xgxdb9 |

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
