## [Editor Report]

This is a very informative and nicely controlled study showing that, when retrieving the value of a food type, bumblebees guide their choices by remembered ranking of feeders instead of focusing on their absolute rewards.

---

## [Decision Letter]

**Decision letter after peer review:**

Thank you for submitting your article "Bumblebees retrieve only the ordinal ranking of foraging options when comparing memories obtained in distinct settings" for consideration by *eLife*.

Your article has been reviewed by three peer reviewers, one of whom is a member of our Board of Reviewing Editors, and the evaluation has been overseen by Christian Rutz as the Senior Editor. The following individual involved in the review of your submission has agreed to reveal their identity: Alex Kacelnik (Reviewer #3).

The reviewers have discussed their reviews with one another, and the Guest Reviewing Editor has drafted this decision letter to help you prepare a revised submission.

Essential revisions:

While all three reviewers found merit in your paper, they raised several important concerns, indicating a need for conducting additional experiments. Here is a summary of the main issues:

Reviewer #1 suggests that the results of the first two experiments could be accounted for by a recency effect, where the higher item of the last comparison is better retrieved, and that in the last experiment, bumblebees' choice for the higher-ranking item could be based on its higher absolute quantitative value in terms of sucrose solution.

Reviewer #2 asks for an experimental assessment of whether bees do respond differently to the different colors and concentrations used. Moreover, Reviewer #2, as well as Reviewer #1, noted that the huge literature on transitive inference in non-human animals could be relevant to the topic addressed by the authors and deserves to be considered.

Reviewer #3, similarly to Reviewer #1, is not convinced that the authors can exclude sensitivity to absolute properties, and suggests some important control experiments, the most important one being the following: training of bees with only A and only B, each in a context, and then giving a choice: if bees show a preference when the only available information is concentration, then the argument that they just cannot remember concentrations could be rejected.

Several other, more specific, comments were provided by the reviewers and should be taken into account by the authors in a revision. In conclusion, the reviewers feel that the paper is potentially publishable but more experimental work is needed to address several issues raised by these fascinating experiments.

Note: Please note that *eLife* has adopted the STRANGE framework, to help improve reporting standards and reproducibility in animal behaviour research. In your revision, please consider scope for sampling biases and potential limitations to the generalisability of your findings:

https://reviewer.elifesciences.org/author-guide/journal-policies

https://doi.org/10.1038/d41586-020-01751-5

*Reviewer #1 (Recommendations for the authors):*

It seems to me that the study is interesting and highlights some fascinating ability of bees to adapt and implement different strategic behaviour to better succeed in different tasks.

*Reviewer #2 (Recommendations for the authors):*

Line 61-63: "Only recently" refers to the work of Pompilio and Kacelnik (1), which is already 12 years old. This does not seem "recent".

Line 63: Please avoid superlative qualifications ("impressive" studies).

Line 70: The name REAL in capitals should be changed to the appropriate citation format.

Lines 118-124: The explanations provided for the lexicographic and the non-lexicographic combinations strategies are not intuitive and difficult to follow both for advised and unadvised readers. Please, be more didactic in your explanations.

Line 200-201: In fact, the work of Greggers and Menzel (1) shows that honey bees can memorize simultaneously four different feeding options providing different reward qualities (sucrose flow rate) and labeled with different colors (and they match their choice to their feeding properties). Mentioning/discussing this would be useful.

Line 205: I could not understand why you are citing here the work of Miriyala et al., (2), which is a work on electrophysiological properties of gustatory receptors with no relation to brain encoding of utility. Miriyala et al., measured peripheral gustatory activity, not central encoding of gustatory information.

References

1. U. Greggers, R. Menzel, Memory dynamics and foraging strategies of honeybees. Behavioral Ecology and Sociobiology 32, 17-29 (1993).

2. A. Miriyala, S. Kessler, F. C. Rind, G. A. Wright, Burst firing in bee gustatory neurons prevents adaptation. Curr Biol 28, 1585-1594 (2018).

*Reviewer #3 (Recommendations for the authors):*

L87: "if bumblebees remembered absolute values for options, their preference between B and C in the test should be identical"

The word 'remembered' should be changed to 'were driven exclusively by their memory for'. An animal can 'remember' something but not use only this memory when other cues are available, as it appears elsewhere in the paper.

L97, Figure 1: It would have been nice for completeness to have a control where bees were trained with A(45) and C(30) in the absence of any competitor flower types, and then offered a choice of A vs. C. I assume that they would prefer A, but if the authors are correct in claiming that bees don't retain absolute memories, even in that case the bees would be indifferent. If the bees in such a control preferred A, this would show that they DO remember quality when other cues are not competing. As it is, the data are compatible with attention to absolute concentration being overshadowed by information about ranking, rather than bees being unable to remember absolute values.

L130: "If bumblebees have memories for both ranking and absolute information, they should either prefer B over C because it is much higher in quality, or this value difference would cancel out the ranking difference (B < A while C > D) and their preference for B and C should be equal. On the other hand, a preference for the much lower reward quality flower C would indicate they have access to only ranking information."

This logic is not completely tight. If, say, the difference in preference induced by absolute memories were X, and that induced by ranking were Y, where X <

L142: "remembered ranking would predict a similar preference. There was no difference in preference for C across experiments 1 and 2 (GLM: 95% CI = [-0.24 0.13], N = 80, P = 0.55; Figure 1B and D)."

From visual inspection of Figure 1B and D it appears that there is in fact a small (and noisy) greater preference in 1B than in 1C. The analysis presented shows that this is not statistically significant. The authors interpret that lack of significance as meaning lack of a difference. This should be tempered, for the usual reasons: lack of significance is not enough to confirm that the null hypothesis (lack of effect) is actually true.

L162 "Further, bumblebees' equal preference for the equally ranked A and C flowers, despite their difference in sugar concentration, demonstrates absolute memories were not used (Table 1) and, in combination with experiments 1 and 2, confirms that bumblebees only use remembered ranking."

I am not convinced. Say that an animal encounters flowers A and B in a random sequence. It then assigns a value to each according to their ranking in that context (A > B), regardless of their absolute concentration. But, each time the animal encounters B, how does it know that it is worse than A? It must be because it DOES remember A, otherwise it could not establish a ranking. In other words, bees may not use, or may not remember, absolute values when moving to new contexts, but they must be sensitive to absolute values, at least within a context. So, maybe there is only short-term memory for absolute value, or maybe that is erased (reset) whenever a new context is entered, but the task of ranking is impossible without some memory for concentration existing and being used.

Again, the simple control of training with only A and only B, each in a context, and then giving a choice would be highly informative: if they show a preference when the only available information is concentration, then the argument that bumblebees just cannot remember concentrations could be rejected.

L182 "In the unrewarding test with all four options available, bees preferred option A (GLM: 95% CI = [0.94,1.68], N = 20, P = 1.21e-06; Figure 2B), indicating that despite being susceptible to suboptimal outcomes in lab-created situations, using only memories of options' ordinal ranking may be evolutionarily rational, i.e. can still lead to optimal choices within an ecologically relevant scenario."

I don't think the language here is used tightly. What Exp 4 shows is that while in 2-way choices bees showed strong context effects (Figure 1B), in 4-way choices animals' preferences match the use of absolute memory as well as ranking. Exp 4 is presented as if the only difference with the earlier design were the training condition, but it also differs in the choice procedure (4 vs. 2 options). It is known that other animals are sensitive to the number of options at choice time. The claim that the situation in Exp 4 is more ecologically relevant than that in the previous experiments is not fully explained, and it would be good to make it more convincing. As it is, it appears somewhat post-hoc.

As elsewhere in the paper, the use of the words optimal and sub-optimal is a bit loose. Any partial preference is suboptimal, because an 'optimal' animal (which is a theoretical construct) allocates all its behaviour to the best option in the context. In this article, following a practice in many experimental psychology writings, the words are used to mean any bias in the favourable direction, regardless of its magnitude. How would one call a theoretical bee that showed exclusive preference for the best option, 'super-optimal', perhaps? By definition, it should not be possible to better optimal behaviour. One could simply say that the psychology of the animals may be adaptive under natural conditions, not 'optimal'. In truth, I even don't like this, because we can't test whether animals are well designed or not. Optimality studies test models, not the notion that animals are well adapted.

L279: "The training was deemed complete when either a bee performed 200 landings or when a bee only landed on the mostly rewarded option for three consecutive bouts."

This method could produce an artefactual bias. If a bee did 3 bouts in a flower other than the 'most rewarded', then the test continues, but if it does it in the richest one, the test stops. This traps good results that could occur at random in any sequence. Under the null hypothesis that bees have no preference, this would show a bias towards the most rewarded flower because the data are censored when a good response is shown. It is equivalent to keep adding individuals to a sample until a significant result is observed, and then stop. I assume that bees may never have shown 3 bouts on a lesser flower, and if this is the case, please state so.

---

## [Author Response]

Essential revisions:While all three reviewers found merit in your paper, they raised several important concerns, indicating a need for conducting additional experiments. Here is a summary of the main issues:Reviewer #1 suggests that the results of the first two experiments could be accounted for by a recency effect, where the higher item of the last comparison is better retrieved, and that in the last experiment, bumblebees' choice for the higher-ranking item could be based on its higher absolute quantitative value in terms of sucrose solution.

In the first three experiments, the order in which the pairs of different sucrose concentrations were used across training sessions was counterbalanced (listed in Figure 1— figure supplement 3). Further, the results of our GLMs show that the order of training had no effect on preferences. Therefore, the results of the first two experiments cannot be explained by a recency effect. We have highlighted these points on lines 101 – 105.

As we alluded to in our original submission, and as mentioned by Reviewer #3, bees must be able to assess the absolute properties for at least some short amount of time, otherwise they could not even discriminate sequentially visited flowers. Our work shows that bumblebees either do not retain memory for (or do not utilise encoded) absolute information for very long. We have now made these points clearer in our revised manuscript.

With the addition of new data from experiments which address Reviewer #3’s concerns, we have also provided a clearer rationale and interpretation of experiment 6 (last experiment in our original submission). Please see our responses to Reviewers below for more detail.

Reviewer #2 asks for an experimental assessment of whether bees do respond differently to the different colors and concentrations used.

In our revised manuscript, we now provide support through previous works and our own experiments (original and newly added) that bumblebees can easily discriminate between the colours used and between the sugar concentrations used in our experiments on lines 348 – 353. Please see our response to Reviewer #2’s comment #1 for more detail.

Moreover, Reviewer #2, as well as Reviewer #1, noted that the huge literature on transitive inference in non-human animals could be relevant to the topic addressed by the authors and deserves to be considered.

Transitive inference (TI) is only superficially related to our question and methods applied here. In short, TI tests for the ability to infer relative ranking and requires subjects not to learn absolute information. We tested whether and to what extent bumblebees remember absolute and/or relative ranking information. Nonetheless, we agree that it may be useful to mention briefly how and why these questions and designs are different. Therefore, we have now included a paragraph for this purpose in the Introduction on lines 73 – 89. Please also see our response to Reviewer #2's comment #3 below for more details.

Reviewer #3, similarly to Reviewer #1, is not convinced that the authors can exclude sensitivity to absolute properties, and suggests some important control experiments, the most important one being the following: training of bees with only A and only B, each in a context, and then giving a choice: if bees show a preference when the only available information is concentration, then the argument that they just cannot remember concentrations could be rejected.

This was a great suggestion. We have done the experiment and have shown that bees form no preference when ranking information was unavailable. We discuss the details of this experiment in our response to Reviewer #3 below and in the revised manuscript on lines 204 – 233, and 385 – 401.

Several other, more specific, comments were provided by the reviewers and should be taken into account by the authors in a revision. In conclusion, the reviewers feel that the paper is potentially publishable but more experimental work is needed to address several issues raised by these fascinating experiments.

We have also addressed all specific comments below and hope that you and all reviewers find the revised manuscript acceptable for publication.

Note: Please note that eLife has adopted the STRANGE framework, to help improve reporting standards and reproducibility in animal behaviour research. In your revision, please consider scope for sampling biases and potential limitations to the generalisability of your findings:https://reviewer.elifesciences.org/author-guide/journal-policieshttps://doi.org/10.1038/d41586-020-01751-5Reviewer #1 (Recommendations for the authors):It seems to me that the study is interesting and highlights some fascinating ability of bees to adapt and implement different strategic behaviour to better succeed in different tasks.

Thank you!

Reviewer #2 (Recommendations for the authors):Line 61-63: "Only recently" refers to the work of Pompilio and Kacelnik (1), which is already 12 years old. This does not seem "recent".

Fair point. We wanted to contrast these works (Pompilio and Kacelnik 2010 as well as Bavard et al., 2021 and 2018, and Klein et al., 2017) with those looking at contextual effects, which began 20 and more years ago. To better reflect our intended meaning, we therefore now state “Only more recently”.

Line 63: Please avoid superlative qualifications ("impressive" studies).

Yes, agreed, deleted.

Line 70: The name REAL in capitals should be changed to the appropriate citation format.

Fixed.

Lines 118-124: The explanations provided for the lexicographic and the non-lexicographic combinations strategies are not intuitive and difficult to follow both for advised and unadvised readers. Please, be more didactic in your explanations.

We have now provided a clearer explanation for all the strategies on lines 143 – 162.

Line 200-201: In fact, the work of Greggers and Menzel (15) shows that honey bees can memorize simultaneously four different feeding options providing different reward qualities (sucrose flow rate) and labeled with different colors (and they match their choice to their feeding properties). Mentioning/discussing this would be useful.

We have now included reference of this study in the Discussion on lines 279 – 282. Note that the Greggers and Menzel study, in which four feeding stations were simultaneously present in close proximity to each other, was not designed to determine whether bees remember absolute versus ranking information. But it is similar to our experiment 6 in that it provides a multicontextual semi-realistic foraging situation where the flowers can be visited quickly but cannot be seen together all at once.

Line 205: I could not understand why you are citing here the work of Miriyala et al., (16), which is a work on electrophysiological properties of gustatory receptors with no relation to brain encoding of utility. Miriyala et al., measured peripheral gustatory activity, not central encoding of gustatory information.

We apologise for the confusion, and thank you for highlighting this. We meant to cite Miriyala et al., to point out that information on fine sugar concentration differences are encoded in the peripheral gustatory neural responses. We now have made this clear on lines 302 – 305.

Reviewer #3 (Recommendations for the authors):L87: "if bumblebees remembered absolute values for options, their preference between B and C in the test should be identical"The word 'remembered' should be changed to 'were driven exclusively by their memory for'. An animal can 'remember' something but not use only this memory when other cues are available, as it appears elsewhere in the paper.

True, thanks. Changed.

L97, Figure 1: It would have been nice for completeness to have a control where bees were trained with A(45) and C(30) in the absence of any competitor flower types, and then offered a choice of A vs. C. I assume that they would prefer A, but if the authors are correct in claiming that bees don't retain absolute memories, even in that case the bees would be indifferent. If the bees in such a control preferred A, this would show that they DO remember quality when other cues are not competing. As it is, the data are compatible with attention to absolute concentration being overshadowed by information about ranking, rather than bees being unable to remember absolute values.

Thank you again for this suggestion! As mentioned above in response to your other comments, our new experiments show that when ranking information is absent, i.e. bees were trained on one flower at a time, bees were unable to retain or utilise memories for absolute information beyond the range of their short-term memories (for a few minutes but much less than one hour). Together, now with the help of the new experiments 4 and 5, our results show that bumblebees only retain ordinal ranking information in the long term and for use in novel contexts, and unless the temporal separation between flower experiences is relatively short, bumblebees are unable to use absolute information to even rank flowers. We include description and discussion of the additional experiments 4 and 5 on lines 204 – 233, 385 – 401, and in Figure 2.

L130: "If bumblebees have memories for both ranking and absolute information, they should either prefer B over C because it is much higher in quality, or this value difference would cancel out the ranking difference (B < A while C > D) and their preference for B and C should be equal. On the other hand, a preference for the much lower reward quality flower C would indicate they have access to only ranking information."This logic is not completely tight. If, say, the difference in preference induced by absolute memories were X, and that induced by ranking were Y, where X <

We agree the logic was not totally solid, therefore we have changed the first sentence to the following (on lines 168 – 173):

“If bumblebees retain and use memories for both ranking and absolute information (and the value weights for these two different types of information are not extremely unbalanced), bees should either prefer B over C because it is much higher in quality, or this value difference would cancel out the ranking difference (B < A while C > D) and their preference for B and C should be equal.”

L142: "remembered ranking would predict a similar preference. There was no difference in preference for C across experiments 1 and 2 (GLM: 95% CI = [-0.24 0.13], N = 80, P = 0.55; Figure 1B and D)."From visual inspection of Figure 1B and D it appears that there is in fact a small (and noisy) greater preference in 1B than in 1C. The analysis presented shows that this is not statistically significant. The authors interpret that lack of significance as meaning lack of a difference. This should be tempered, for the usual reasons: lack of significance is not enough to confirm that the null hypothesis (lack of effect) is actually true.

Fair point. We have now tempered this section by changing the wording on lines (181 – 185).

L162 "Further, bumblebees' equal preference for the equally ranked A and C flowers, despite their difference in sugar concentration, demonstrates absolute memories were not used (Table 1) and, in combination with experiments 1 and 2, confirms that bumblebees only use remembered ranking."I am not convinced. Say that an animal encounters flowers A and B in a random sequence. It then assigns a value to each according to their ranking in that context (A > B), regardless of their absolute concentration. But, each time the animal encounters B, how does it know that it is worse than A? It must be because it DOES remember A, otherwise it could not establish a ranking. In other words, bees may not use, or may not remember, absolute values when moving to new contexts, but they must be sensitive to absolute values, at least within a context. So, maybe there is only short-term memory for absolute value, or maybe that is erased (reset) whenever a new context is entered, but the task of ranking is impossible without some memory for concentration existing and being used.Again, the simple control of training with only A and only B, each in a context, and then giving a choice would be highly informative: if they show a preference when the only available information is concentration, then the argument that bumblebees just cannot remember concentrations could be rejected.

We apologise for the confusion, as we were not trying to claim that bumblebees are completely insensitive to absolute information. As you rightly point out, this would make comparing sequentially visited flowers with different sugar concentrations impossible to compare. We now point this out explicitly in our revised manuscript, both in the Abstract and the Results.

Thank you again for the suggested experiment. As we replied above, we now provide data from additional experiments showing that when learning only one flower at a time, bees only retain (or utilise memory of) absolute information for a short amount of time (at least a few minutes but much less than an hour). We now include a description of these experiments on lines 204 – 233, 385 – 401 and in Figure 2. L182 "In the unrewarding test with all four options available, bees preferred option A (GLM: 95% CI = [0.94,1.68], N = 20, P = 1.21e-06; Figure 2B), indicating that despite being susceptible to suboptimal outcomes in lab-created situations, using only memories of options' ordinal ranking may be evolutionarily rational, i.e. can still lead to optimal choices within an ecologically relevant scenario."

I don't think the language here is used tightly. What Exp 4 shows is that while in 2-way choices bees showed strong context effects (Figure 1B), in 4-way choices animals' preferences match the use of absolute memory as well as ranking. Exp 4 is presented as if the only difference with the earlier design were the training condition, but it also differs in the choice procedure (4 vs. 2 options). It is known that other animals are sensitive to the number of options at choice time. The claim that the situation in Exp 4 is more ecologically relevant than that in the previous experiments is not fully explained, and it would be good to make it more convincing. As it is, it appears somewhat post-hoc.

We apologise for the lack of explanation. Our intent was to simply determine whether the presumed strategy taken from the results of experiments 1-3 would lead to bees preferring flowers with lower sugar concentrations when presented in a situation more like they would experience in the wild. Bees forage in situations where different flowers can be seen in their visual field at the same time, but they can and will visit flowers which cannot be seen together with the other visted flowers (i.e. on the opposite side of a bush, or separated by many meters). Essentially, the design of our experiment 6 (previously experiment 4) mimics wild situations where bees drink from various flowers in different spatial locations whereby they cannot see all the different flowers at the same time but the duration between visits to different flowers is short (several seconds to a few minutes). We now explain this in more detail on lines 249 – 261.

With regards to the interpretation of experiment 6 (previously experiment 4), we believe that the addition of the new data from our experiments which address your comments above (now experiments 4 and 5) clearly indicate that bees’ preference differences between experiments 1-3 compared to experiment 6 are due to temporal differences between flower visits. As you rightly mentioned in your other comments, we now highlight more clearly in the revised manuscript that bumblebees must use absolute information to compare, and rank, any two sequentially visited flowers. However, our results suggest that bumblebees either do not retain or utilise absolute information beyond the range of their short-term memories. We now address this more clearly and with additional data on lines 204 – 233.

As elsewhere in the paper, the use of the words optimal and sub-optimal is a bit loose. Any partial preference is suboptimal, because an 'optimal' animal (which is a theoretical construct) allocates all its behaviour to the best option in the context. In this article, following a practice in many experimental psychology writings, the words are used to mean any bias in the favourable direction, regardless of its magnitude. How would one call a theoretical bee that showed exclusive preference for the best option, 'super-optimal', perhaps? By definition, it should not be possible to better optimal behaviour. One could simply say that the psychology of the animals may be adaptive under natural conditions, not 'optimal'. In truth, I even don't like this, because we can't test whether animals are well designed or not. Optimality studies test models, not the notion that animals are well adapted.

We agree and have now removed the terms optimality/suboptimality. We now use, where appropriate, advantageous/disadvantageous.

L279: "The training was deemed complete when either a bee performed 200 landings or when a bee only landed on the mostly rewarded option for three consecutive bouts."This method could produce an artefactual bias. If a bee did 3 bouts in a flower other than the 'most rewarded', then the test continues, but if it does it in the richest one, the test stops. This traps good results that could occur at random in any sequence. Under the null hypothesis that bees have no preference, this would show a bias towards the most rewarded flower because the data are censored when a good response is shown. It is equivalent to keep adding individuals to a sample until a significant result is observed, and then stop. I assume that bees may never have shown 3 bouts on a lesser flower, and if this is the case, please state so.

Bees never went three bouts or more on the more poorly rewarding flower. We now state this in the Methods section on lines 413 – 415.